# Generative Adversarial Optimization: Dual-Reward Reinforcement Learning for Mathematics Reasoning

## Abstract

Despite recent progress achieved by large language models (LLMs), their remarkable mathematics reasoning abilities are largely dependent on fine-tuning on the annotated data, lacking generalization on out-of-distribution tasks. To address this, current methods adopt reinforcement learning (RL) to incentivize the latent capabilities of LLMs, mitigating the need for annotations. However, they often suffer from uncontrollable data difficulty and limited initial capabilities. In this paper, we propose **G**enerative **A**dversarial **O**ptimization (**GAO**), a novel reinforcement learning framework consists of a problem poser and a problem solver which are optimized by dual-reward iteratively. Specifically, the poser attempts to propose challenging problems to stump the solver, while the solver strives to solve them. The complete adversarial process is recorded to generate bidirectional rewards, enabling both the poser and solver to co-evolve through this competitive interaction. Experimental results show that **GAO** achieves state-of-the-art performance compared to previous models of the same size, even without relying on proprietary LLMs.

## 1 Introduction

Recent large language models (LLMs) have demonstrated impressive performance on reasoning tasks (Wang et al., 2023; Taylor et al., 2022). These successes highlight that fine-tuning on vast amounts of annotated data significantly enhances their core reasoning abilities. However, the effectiveness of post-training is heavily dependent on the availability of high-quality annotated data (Yu et al., 2024; Yuan et al., 2023; Luo et al., 2025; Li et al., 2024), and challenges of data collection and annotation remain difficult to overcome (Feng et al., 2025). Given that, some approaches like Qwen3 (Yang et al., 2025), Seed-Thinking-v1.5 (Yu et al., 2025b), Light-R1 (Wen et al., 2025), and DeepSeek-R1 (DeepSeek-AI et al., 2025), adopt reinforcement learning (RL) to mitigate the need for annotations. They encourage LLMs to explore the output space and use rewards to reinforce or penalize the behavior of LLMs, leading to better generalization (Chu et al., 2025). However, the difficulty of training data is uncontrollable, which makes the effect of RL hard to guarantee. Too complex or too easy inputs usually contribute to highly skewed reward distribution (Team et al., 2025), significantly diminishing learning efficiency. Besides, the initial capabilities of LLMs can also affect the ceiling of RL (AI et al., 2025). The capabilities of a model trained through reinforcement learning do not grow indefinitely (Gandhi et al., 2025). For certain challenging problems, LLMs often fails to find the optimal solution even after multiple explorations (Yue et al., 2025), leading to the failure of RL, and thus the performance of models may stagnate.

The challenges mentioned above motivated us to propose **G**enerative **A**dversarial **O**ptimization (**GAO**), which enhances the reasoning abilities of LLMs through dual-reward reinforcement learning. As illustrated in Figure 1, SFT trains LLMs to memorize the training data, often struggling to generalize to out-of-distribution tasks, while RL employs reward models (or reward functions) to provide feedback during the exploration process of the model, yet still performs poorly in difficult scenarios having a gap with the model's capabilities as the exploration in such areas is often insufficient. In contrast, **GAO** progressively enhances the model's capabilities by continuously identifying and targeting its weaknesses. Specifically, we first utilize a problem poser to construct challenging questions which the problem solver (i. e. the target reasoning model) can hardly answer, finding the

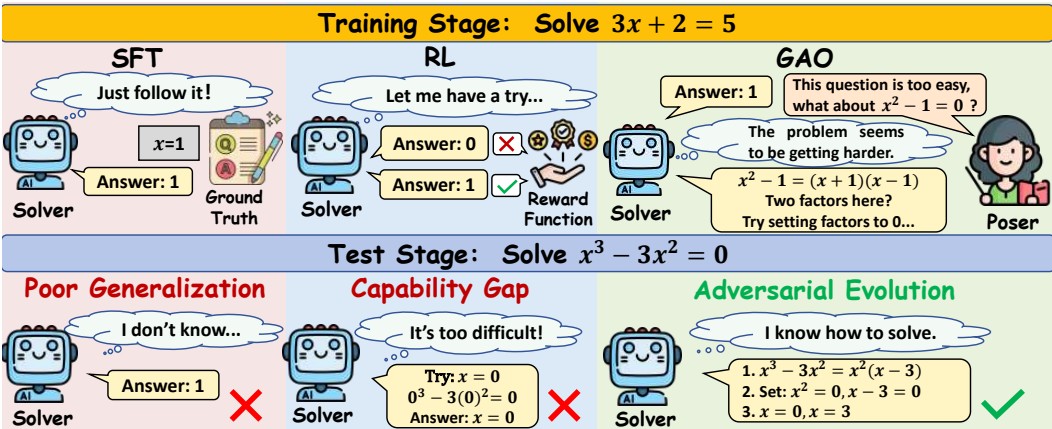

Figure 1: The comparisons between our method and previous training methods. SFT trains LLMs to fit the training data, contributing to the poor generalization. Traditional reinforcement learning methods (e.g., PPO, GRPO, and so on) encourage LLMs to explore different solutions and utilize reward functions to provide feedback, leading to failure when the problems exceed the models' capabilities. In contrast, GAO progressively enhances the model's capabilities by continuously identifying and targeting its weaknesses.

tasks where the target model can obtain more gains. Then we conduct RL for the solver on these questions with consistency voting from external reasoning experts. Meanwhile, we also train and update the poser with the reward signal derived from solver's performance on generated questions. The process of **GAO** mentioned above proceeds iteratively. Through this process, the poser becomes increasingly tricky, while the solver grows progressively strong and robust. The main contributions of this paper are summarized as follows:

- We identify the limitations of current post-training methods (SFT and RL) and design a novel training paradigm to overcome them.

- We propose **G**enerative **A**dversarial **O**ptimization (**GAO**), which consists of a poser and a solver and enhances the reasoning abilities of LLMs through dual-reward reinforcement learning. Compared to the previous methods on the backbones of the same size, **GAO** improves the ceiling of the post-training, achieving state-of-the-art (SOTA) performance.

- Extensive experiments demonstrate the effect of **GAO** on multiple reasoning-related tasks, with ablation and analysis studies explaining how and why it works.

## 2 RELATED WORK

### 2.1 REASONING LARGE LANGUAGE MODEL

Reasoning plays a crucial role in intellectual activities of LLMs (Huang & Chang, 2023), attracting significant interest from both academia and industry. With the increase of the size, recent LLMs have made significant advancements in a wide range of reasoning tasks such as arithmetic, commonsense, symbolic reasoning (Qiao et al., 2023). Despite such impressive progress, the performance of current open-source reasoning LLMs (Wen et al., 2025; DeepSeek-AI et al., 2025; Yang et al., 2025) still lags behind that of proprietary ones (e.g., o3, o4-mini, gemini-2.5-pro, gpt-4.5-preview, etc. (Anil et al., 2023; Ouyang et al., 2022; OpenAI et al., 2024)) (Chiang et al., 2024), primarily because stronger models often keep their training data proprietary. As a result, the lack of publicly available reasoning datasets remains a significant barrier to further development in this field.

### 2.2 GENERATIVE ADVERSARIAL NETWORK

Generative Adversarial Network (GAN) is a deep network designed for generating synthetic data that mimics real data distributions. Introduced by Goodfellow et al. (2020), GAN consists of a gen-

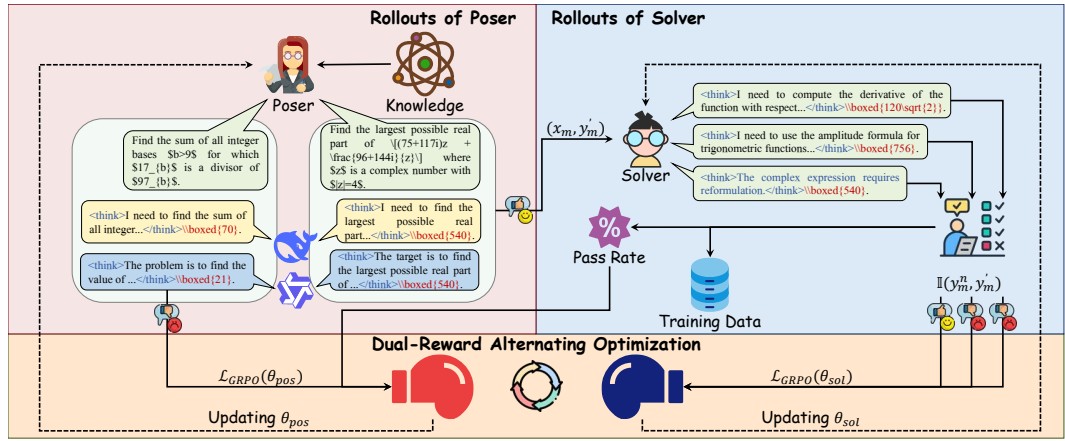

Figure 2: Illustration of the GAO training paradigm. At each turn, the Poser samples candidate problems, which are filtered through a consistency check before being passed to the Solver. The Solver's pass rate not only guides its own training but also provides feedback for the Poser to generate increasingly challenging problems.

erator creating fake data samples and a discriminator distinguishing between real and fake samples. Through such adversarial training (Gulrajani et al., 2017), GAN achieves high-fidelity data generation across various domains, including images, text, and audio. Inspired by GANs, recent works in code generation and mathematical reasoning, such as Absolute-Zero-Reasoner (Zhao et al., 2025) and SvS (Liang et al., 2025), enable self-play by applying both problem-generation and problem-solving objectives to a single model. However, this design leads to unstable training, and a more effective solution still needs to be explored.

### 2.3 DATA FILTERING FOR REINFORCEMENT LEARNING

The quality of training data is crucial for the final performance of Reinforcement Learning. The problems like noisy or sparse rewards (Hare, 2019), shifted distributions (Agarwal et al., 2021), exploration bottlenecks (Bai et al., 2021), and adversarial trajectories (Pinto et al., 2017) can significantly affect its effectiveness. Therefore, the process of selectively curating or preprocessing training data to enhance the efficiency and robustness of RL sampling, bridging gaps between simulated and real-world deployment (Kumar et al., 2020).

## 3 METHODOLOGY

In this section, we describe how **GAO** progressively enhances the model's capabilities through dual-reward reinforcement learning. Specifically, we first initialize the poser and solver with SFT datasets, then we optimize both of them in a generative adversarial way. The training process of the poser and solver iteratively and alternately proceeds until no more gain can be obtained. It is worth noting that we treat the output of their interaction as the training trajectory of **GAO**, replacing model updating with data updating to avoid severe shifted distribution issues.

### 3.1 INITIALIZATION FOR THE POSER AND SOLVER

The basic capabilities of the models are crucial for subsequent reinforcement learning. Given that, we initialize the solver by fine-tuning it on the open-source SFT dataset - DeepMath-103K (He et al., 2025). The training objective can be formulated as:

$$\mathcal{L}_{\text{SFT}}(\theta_{sol}) = -\sum \log P_{\theta_{sol}}(y^* \mid x) \tag{1}$$

where $\theta_{sol}$ indicates the parameter of the solver. $x$ and $y^*$ are the input (question) and gold output (solution).

As for the poser, we hope it can design various questions based on different knowledge points and ultimately construct high-quality, diverse data. Inspired by PromptCoT (Yao et al., 2024), we utilize Qwen2.5-72B-Instruct (Team, 2024) to summarize knowledge points $k$ which are used by the poser to generate questions (see Appendix A for details). After collecting $k$, we can initialize the poser:

$$\mathcal{L}_{\text{SFT}}(\theta_{pos}) = -\sum \log P_{\theta_{pos}}(c, x \mid k) \tag{2}$$

where $\theta_{pos}$ indicates the parameter of the poser. $c$ can be regarded as the Chain-of-Thoughts before formulating the final questions.

## 3.2 Training for the Poser

Each training iteration of **GAO** starts with the poser's training phase. The poser is expected to identify the weaknesses of the solver, so we use the failure rates of the solver on the synthetic problems generated by the poser as the rewards to optimize it. Specifically, we first utilize the poser to sample $M$ questions $x_{1:M}$ based on each knowledge point $k$. Then we employ strong reasoning models (Qwen3-235B-A22B (Yang et al., 2025) and DeepSeek-R1 (DeepSeek-AI et al., 2025)) to conduct consistency voting, getting the top-voted answer $y'_m$ for each question $x_m$. Given that, we can get the reward function for the poser:

$$x_1, x_2, ..., x_M \sim \pi_{\theta_{pos}}(k) \quad y_m^1, y_m^2, ..., y_m^N \sim \pi_{\theta_{sol}}(x_m)$$

$$r_m^{pos} = \begin{cases} 1 - \frac{1}{N} \sum_{n=1}^N \mathbb{I}(y_m^n = y'_m) & \exists n, \mathbb{I}(y_m^n, y'_m) = 1 \\ 0 & \forall n, \mathbb{I}(y_m^n, y'_m) = 0 \end{cases} \tag{3}$$

where $N$ is the number of rollouts of the solver and $\mathbb{I}$ is the rule-based judge function. $y_m^n$ is the $n$-th output sampled from the solver fed with the question $x_m$.

From the perspective of the overall training objectives, the poser serves as an indicator, highlighting the gap between the solver and the current state-of-the-art reasoning model. In particular, when these strong models fail to reach a consensus or the solver is completely unable to correctly answer a question posed by the poser, we consider that the question is overly difficult or potentially unsolvable. Such problems provide no benefit, or may even be harmful, to the subsequent RL for the solver. Therefore, we discourage the poser from generating such questions and set their reward to 0. Finally, we employ GRPO to optimize the poser. Inspired by DAPO (Yu et al., 2025a) and Dr.GRPO (Liu et al., 2025), we remove the KL term and adopt token-level average pooling as the loss aggregating mode:

$$\mathcal{L}_{\text{GRPO}}(\theta_{pos}) = \mathbb{E}_{k \sim \mathcal{K}, x \sim \pi_{\theta_{pos}}(\cdot|k)} \frac{1}{\sum_{m=1}^M |x_m|}$$

$$\left[ \sum_{m=1}^M \sum_{t=1}^{|x_m|} (\min(z_m^t A_m^t, \text{clip}\left(z_m^t, 1-\varepsilon, 1+\varepsilon\right) A_m^t)) \right] \tag{4}$$

$$z_m^t = \frac{\pi_{\theta_{pos}}(x_{m,t} \mid k, x_{m,<t})}{\pi_{\theta_{pos}^{old}}(x_{m,t} \mid k, x_{m,<t})}, A_m^t = \frac{r_m^{pos} - avg(r_{1:M}^{pos})}{std(r_{1:M}^{pos})}$$

## 3.3 Training for the Solver

The training process for the solver is conducted on the poser-generated problems. Similar to the sampling process during the training stage of the poser, we can collect problems with top-voted answers $y'_m$. Given that, we can obtain the reward function for the solver:

$$r_n^{sol} = \mathbb{I}(y_m^n, y'_m) \tag{5}$$

Before the prior to the formal training phase, we conduct problems filtering to select samples with a pass rate between $P_{low}$ and $P_{high}$. Such data presents an appropriate level of challenge to the solver's reasoning capabilities, which yields greater benefits during reinforcement learning. On the other hand, the filtering process can significantly reduce the volume of training data, thereby greatly improving training efficiency.

Based on the filtered data, we fine-tune the solver's policy to maximize rewards, encouraging it to make precise, step-by-step logical deductions. The training objective is defined as:

$$\mathcal{L}_{\text{GRPO}}(\theta_{sol}) = \mathbb{E}_{x_m \sim \pi_{\theta_{\text{pos}}}, y_m^n \sim \pi_{\theta_{\text{sol}}}(\cdot | x_m)} \frac{1}{\sum_{n=1}^{N} |y_m^n|}$$

$$\left[ \sum_{n=1}^{N} \sum_{t=1}^{|y_m^n|} (\min(z_n^t A_n^t, \text{clip}\left(z_n^t, 1 - \varepsilon, 1 + \varepsilon\right) A_n^t)) \right] \tag{6}$$

$$z_n^t = \frac{\pi_{\theta_{\text{sol}}}(y_m^{n,t} \mid y_m^{n,<t})}{\pi_{\theta_{\text{sol}}^{old}}(y_m^{n,t} \mid y_m^{n,<t})}, A_n^t = \frac{r_n^{sol} - avg(r_{1:N}^{sol})}{std(r_{1:N}^{sol})}$$

### 3.4 DUAL-REWARD ALTERNATING OPTIMIZATION

To ensure the training stability, GAO alternately optimizes the poser and solver, enabling the target model to focus more on the current task in each optimization step. This training approach is inspired by GAN: the poser acts as the generator, striving to identify capability gaps between the solver and SOTA mathematics reasoning LLMs while posing problems; the solver serves as the discriminator, aiming to solve the proposed problems accurately without being stumped by the poser.

---

**Algorithm 1** GAO Algorithm

---

**Input:** Poser $\theta_{pos}^0$, Solver $\theta_{sol}^0$, Initial SFT Datasets $D_{pos}$ and $D_{sol}$
**Output:** Solver $\theta_{sol}$
1: Initialize the parameters of Poser and Solver on $D_{pos}$ and $D_{sol}$ by $\mathcal{L}_{\text{SFT}}$: $\theta_{pos}^0$, $\theta_{sol}^0$
2: Initialize the RL training buffer $B \leftarrow \varnothing$
3: **for** $i = 0$ **to** $T$ **do**
4:     Sample problems from the poser based on the given knowledge points $k$: $\{x_m\}_{m=1}^M \sim \pi_{\theta_{pos}^i}$
5:     Employ strong reasoning models to get the top-voted answer $y_m'$
6:     Sample trajectories from the solver for each $x_m$: $\{y_m^n\}_{n=1}^N \sim \pi_{\theta_{sol}^i}$
7:     Calculate $r_{pos}^m$ in accordance with Equation 3 and then train the poser by $\mathcal{L}_{\text{GRPO}}(\theta_{pos})$: $\theta_{pos}^{i+1} \leftarrow \theta_{pos}^i$
8:     **for** $m = 1$ **to** $M$ **do**
9:         Calculate pass rate $p_m$ and $\{r_m^n\}_{n=1}^N$ for $\{y_m^n\}_{n=1}^N$
10:        **if** $P_{low} \leq p_m \leq P_{high}$ **then**
11:            Append $\{x_m, (y_m^1, r_m^1), ..., (y_m^N, r_m^N)\}$ to $B$
12:        **end if**
13:     **end for**
14:     Update the solver by $\mathcal{L}_{\text{GRPO}}(\theta_{sol})$: $\theta_{sol}^{i+1} \leftarrow \theta_{sol}^i$ using the buffer $B$
15: **end for**
16: **return** $\theta_{sol}^T$

---

As shown in Algorithm 1, GAO alternately updates the poser and solver in an adversarial loop: the poser generates progressively harder problems that expose the solver's weaknesses, while the solver improves by attempting to solve them. The solver's success rate is used as feedback to push the poser to produce harder, more informative tasks. Through this competition, posing and solving improve together.

## 4 EXPERIMENTS

### 4.1 SETTINGS

**Models and Datasets** We adopt Qwen3-8B-Base (Yang et al., 2025) as the backbone for the solver and Qwen3-8B as the backbone for the poser to evaluate the effectiveness of our method. We use the open-source mathematical dataset DeepMath-103K (He et al., 2025) as the SFT dataset for initializing the solver. After applying simple deduplication, we use the resulting SFT dataset

to perform the initialization training of the solver. For the poser initialization, we randomly sample problems from the above datasets and generate SFT data using Qwen2.5-72B-Instruct (Team, 2024), with the prompts detailed in Appendix A.

**Implementation Details**    We employ verl (Sheng et al., 2024) as the reinforcement learning framework, and all experiments are conducted on 8 NVIDIA H20 96G GPUs. During the GAO iterative stage, the number of game rounds is set to $T = 4$. In each round the poser samples $M = 4096$ problems. For each generated problem, the solver's GRPO rollout sampling count is set to $N = 8$. The sampling temperature is set to 1.0, with top_p = 0.7. The thresholds for difficulty-based filtering were set as a pass rate of $P_{low} = 25.0\%$ and $P_{high} = 62.5\%$.

**Baselines**    The baselines consist of basemodels and fine-tuned models.

*Base Models* are pretrained on large-scale general knowledge corpora and serve as foundational pretrained backbones for subsequent fine-tuning on mathematical tasks. The base models include Qwen3-8B-Base,Qwen3-14B-Base (Yang et al., 2025) and Qwen2.5-7B-Base,Qwen2.5-14B-Base (Team, 2024).

*Fine-tuned Models* are adapted from base models through fine-tuning on smaller, domain-specific mathematical datasets. The fine-tuned models include Qwen2.5-Math-7B-Instruct (Team, 2024), SVS Liang et al. (2025),PromptCot (Yao et al., 2024), OpenMath-Nemotron-14B (Moshkov et al., 2025), lightR1 (Wen et al., 2025) and Critique-GRPO (Zhang et al., 2025). In addition, we also evaluate models that are not specifically fine-tuned on mathematical datasets but demonstrate strong general reasoning ability, such as DeepSeek-R1-Distill-Qwen-14B (DeepSeek-AI et al., 2025), MEGA-SCIENCE (Fan et al., 2025),AZR (Zhao et al., 2025) and Qwen3-8B-Instruct (Yang et al., 2025),

**Evaluation**    We evaluate the performance of our model on authoritative mathematics benchmark datasets, including the relatively easy GSM8K (Cobbe et al., 2021), the medium-difficulty MATH-500 (Lightman et al., 2023), and higher-difficulty datasets AIME24, AIME25, and OlymMATH (Sun et al., 2025). To ensure fairness, all model inferences are conducted using the VLLM (Kwon et al., 2023) framework. The inference settings use a temperature of 0.6 and a maximum output length of 65,536 tokens. To reduce the effect of measurement variance, all reported results are averaged over eight runs.

## 4.2 MAIN RESULTS

The performance of **GAO** on mathematical reasoning benchmarks is summarized in Table 1. GAO achieves strong results on benchmarks of varying difficulty, reaching a pass@1 accuracy of **94.6%** on GSM8K and **94.3%** on MATH500, outperforming baseline models. Moreover, GAO demonstrates substantial improvements on high-difficulty benchmarks, achieving **82.1%** on AIME 2024, **67.5%** on AIME 2025, and **57.2%** on OlymMATH, establishing new state-of-the-art results. This pronounced gain is a direct consequence of GAO's iterative game-theoretic training: the poser dynamically generates increasingly challenging problems tailored to the solver's current capabilities, effectively pushing the solver to extend its reasoning depth. As a result, GAO performs well on easier tasks and shows clear improvements on more challenging mathematical problems, illustrating the benefits of adversarial, difficulty-adaptive training.

To ensure a rigorous and objective evaluation, we compared GAO with a set of competitive open-source mathematical reasoning models. As shown in Table 1, GAO performs prominently among 7–8B scale models. On more challenging high-difficulty benchmarks such as AIME24, AIME25, and OlymMATH, GAO demonstrates clear advantages. Notably, compared with other GAN-inspired adversarial training methods like SvS and AZR, GAO can significantly raise the model's capability ceiling, reflecting the superiority of training the poser and solver separately: it avoids the training instability caused by multiple objectives acting on a single model. Compared with the second-best model of similar size, Qwen3-8B, GAO achieves a notable improvement of 6.0% on AIME24 and 7.6% on OlymMATH, highlighting the effectiveness of our method in handling complex mathematical reasoning tasks. We further compared GAO with larger 14B models. GAO outperforms OpenMath-Nemotron-14B, which is carefully optimized on domain-specific mathematical data, across all benchmarks. In particular, GAO surpasses Qwen3-14B on high-difficulty benchmarks, further demonstrating its robustness in challenging scenarios. These results indicate that the iter-

| Models | Backbone | GSM8K | MATH500 | AIME24 | AIME25 | OlymMATH | AVG |
|---|---|---|---|---|---|---|---|
| **Base Models** | | | | | | | |
| Qwen2.5-7B-Base | _ | 82.0 | 53.4 | 12.1 | 0.0 | 4.5 | 30.4 |
| Qwen2.5-14B-Base | _ | 86.5 | 55.6 | 13.3 | 3.3 | 3.3 | 32.4 |
| Qwen3-8B-Base | _ | 57.6 | 52.8 | 15.0 | 7.5 | 4.3 | 27.4 |
| Qwen3-14B-Base | _ | 87.5 | 71.4 | 21.3 | 10.8 | 7.2 | 39.6 |
| **14B Fine-tuned Models** | | | | | | | |
| DS-Distill-14B | Qwen2.5 | 88.7 | 91.0 | 65.0 | 50.0 | 37.5 | 66.4 |
| OpenMath-Nemotron-14B | Qwen2.5 | 93.7 | 94.5 | 65.8 | 49.2 | 38.1 | 68.3 |
| Light-R1-14B | Qwen2.5 | 95.8 | 94.7 | 75.0 | 53.3 | 40.9 | 71.9 |
| Qwen3-14B | Qwen3 | **96.0** | **96.2** | **81.2** | **62.5** | **55.9** | **78.4** |
| **7-8B Fine-tuned Models** | | | | | | | |
| Llama3.1-8B-Instruct | Llama3.1 | 81.3 | 44.0 | 15.8 | 0.8 | 2.5 | 28.8 |
| SvS-8B | Llama3.1 | 90.3 | 62.2 | _ | _ | 26.4 | 59.6 |
| Qwen2.5-Math-7B-Instruct | Qwen2.5 | 94.4 | 83.8 | 14.2 | 10.0 | 7.0 | 41.9 |
| AZR-7B | Qwen2.5 | _ | 72.6 | 20.0 | 10.0 | 38.2 | 35.2 |
| Light-R1-7B | Qwen2.5 | 86.5 | 91.6 | 59.7 | 45.3 | 28.1 | 62.2 |
| PromptCot-7B | Qwen2.5 | 92.8 | 93.7 | 58.7 | 49.2 | 20.5 | 63.0 |
| MEGASCIENCE-8B | Qwen3 | 92.7 | 85.9 | 32.5 | 25.8 | 10.8 | 49.5 |
| Critique-GRPO-8B | Qwen3 | 93.1 | 93.4 | 68.3 | 49.2 | 35.6 | 67.9 |
| Qwen3-8B | Qwen3 | 93.9 | 91.37 | 76.1 | 65.6 | 49.6 | 75.1 |
| **GAO(Ours)** | Qwen3 | **94.6** | **94.3** | **82.1** | **67.5** | **57.2** | **79.1** |

Table 1: Compare the pass@1(%) accuracy of competitive models across various mathematical benchmarks. Highlight boldface values to indicate the best performance within models of the same size, and use underlined values to denote the second-best performance at that szie.

| Dataset | DeepSeek-Distill-7B | |
|---|---|---|
| | Accuracy($\downarrow$) | AVG. Reasoning Tokens($\uparrow$) |
| AIME24 | 55.5 | 4583 |
| LightR1 | 49.7 | 6572 |
| PromptCot | 48.9 | 6305 |
| GAO | **46.6** | **7172** |

Table 2: Problem difficulty comparison across datasets, measured by reasoning trajectory length and pass rate with DeepSeek-R1-Distill-Qwen-7B.

ative adversarial interplay between the poser and solver enables GAO to overcome the limitations of model scale, substantially enhancing the solver's mathematical reasoning ability to reach or even exceed the level of larger models.

## 4.3 ANALYSIS

**Difficulty Analysis**  To measure the difficulty gap between problems generated by Poser and those from other datasets, we randomly sampled up to 300 problems from each dataset and evaluated them using the DeepSeek-Distill-7B model. We quantify problem difficulty using model accuracy, with lower accuracy indicating harder problems, and the average length of reasoning trajectories, with longer trajectories indicating more complex reasoning.As shown in Table 2, GAO generates the most challenging problems: it has the lowest model accuracy (46.6%) and the longest average reasoning trajectories (7172 tokens). By comparison, AIME24 problems are easier, with 55.5% accuracy and 4583 reasoning tokens, while LightR1 and PromptCot lie in between. This result demonstrates GAO's effectiveness in the adversarial iterative process, producing a greater number of high-difficulty, reasoning-intensive problems.

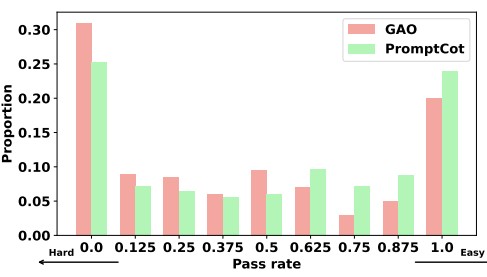 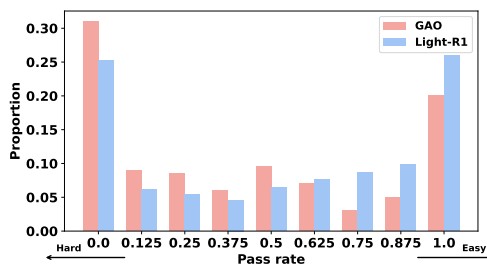

(a) PromptCot vs. GAO Difficulty Distribution      (b) Light-R1 vs. GAO Difficulty Distribution

Figure 3: Pass rate distribution of generated problems. GAO produces more difficult and high-value problems, providing richer training signals for solver fine-tuning.

In addition, to investigate whether the generated problems are suitable for training the solver, we employ the model trained from GAO to evaluate both the problems generated by PromptCot, Light-R1 and those generated by GAO. We randomly sample 300 problems from each source and perform 8 inference runs for each problem, then analyze the distribution of samples across different pass rate intervals. Intuitively, if a larger proportion of problems fall into lower pass rate intervals, this indicates higher difficulty. If a larger proportion lies between 25% and 62.5%, it suggests that the generated problems have moderate difficulty relative to the current reasoning model, making them more suitable for reinforcement learning training. Figure 3(a) and Figure 3(b) compare the difficulty distributions of GAO-generated problems with those from the PromptCot and Light-R1 datasets. GAO-generated problems are more heavily concentrated near a pass rate of 0, highlighting their increased difficulty. Moreover, a notable fraction of problems falls within the 25%–62.5% pass rate range, suggesting that GAO's poser generates a larger proportion of high-value problems. This distribution offers richer training signals, thereby providing more effective guidance for reinforcement learning optimization.

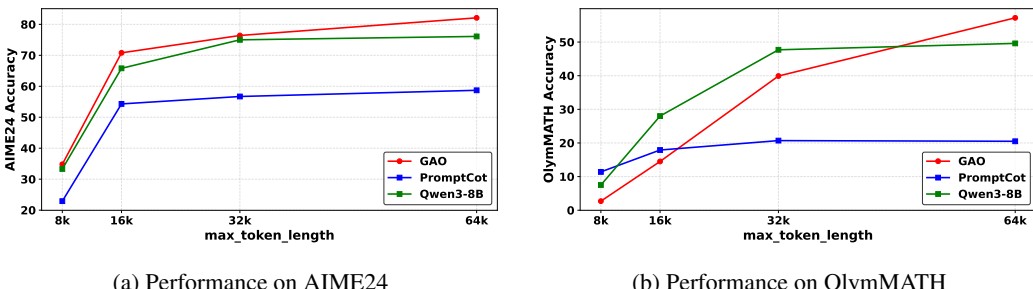

(a) Performance on AIME24      (b) Performance on OlymMATH

Figure 4: Effect of token budget on GAO's chain-of-thought reasoning and problem-solving performance on AIME24 and OlymMATH.

**Reasoning Ability Analysis** Figure 4 illustrates that on the AIME24 and OlymMATH benchmark. As the token budget increases, our GAO model is able to more effectively leverage chain-of-thought (CoT) reasoning, thereby achieving stronger problem-solving performance. Recent works in mathematical reasoning tasks, such as PromptCoT, DeepSeek-R1, and Qwen3, have highlighted the critical role of extending the chain-of-thought in solving challenging problems, and have emphasized the importance of CoT length and quality during dataset construction to enhance reasoning capabilities. From the results in Figure 4(b), we observe that GAO is capable of autonomously extending its CoT length on harder problems, thereby lengthening its reasoning process. Increasing the token budget not only prevents premature truncation due to budget constraints but also ensures the completeness of the reasoning chain, allowing the model's reasoning capacity to be fully utilized.

In Figure 4(b), we observe pronounced differences in scaling behavior on high-difficulty problems. While Qwen3-8B and PromptCot begin to plateau as the token budget nears 64k, GAO shows a much milder tendency toward saturation, with performance still improving in the large-context regime.

This pattern shows that GAO effectively uses extra tokens to form longer, coherent reasoning chains, yielding more accurate solutions. Unlike baselines, it converts extended context into consistent performance gains, highlighting its ability to exploit long-context information for reliable problem solving.

**Iterative Game Dynamics**    To evaluate the effectiveness of iterative adversarial training, we examine solver performance across successive rounds of poser–solver interactions. We conduct four rounds so that the poser can progressively generate more challenging problems, while the solver has ample opportunity to refine its reasoning strategies. By repeatedly engaging both components in this game-like setup, the poser incrementally escalates problem difficulty, while the solver is continuously driven to refine its reasoning ability.

| $Turn$ | AIME24 | AIME25 | OlymMATH | AVG |
|:---:|:---:|:---:|:---:|:---:|
| 0 | 74.8 | 60.3 | 47.5 | 60.9 |
| 1 | 77.3 | 62.5 | 49.3 | 63.0 |
| 2 | 79.4 | 64.5 | 55.3 | 66.4 |
| 3 | 81.4 | 66.9 | 56.7 | 68.3 |
| 4 | **82.1** | **67.5** | **57.2** | **68.9** |

Table 3: Effect of game iterations on solver performance. Accuracy improves consistently with more iterations, especially on harder benchmarks.

Table 3 reports solver accuracy on AIME24, AIME25, and OlymMATH for different numbers of game iterations. $Turn$ refers to the number of adversarial rounds between the Poser and the Solver. We observe a clear upward trend: performance increases steadily from 60.9% at initialization to 68.9% after four iterations. Notably, the largest relative gain appears on the more difficult Olym-MATH (+9.7 points), suggesting that iterative interactions help with harder problems.

Although accuracy improves consistently, the performance gains begin to taper after the third turn, indicating diminishing returns with further iterations. Based on this observation, we limit the number of rounds to four to balance improvements with computational cost. These results highlight that iterative poser–solver dynamics not only produce steady gains across benchmarks but also offer the most pronounced benefits for tasks requiring deeper reasoning, thereby enhancing both robustness and generalization of the solver.

## 5    CONCLUSION, LIMITATION AND FUTURE WORKS

Current mathematical reasoning solvers often hit a performance ceiling during iterative training, primarily due to training data that is either insufficiently challenging or poorly aligned with the solver's capabilities. To address this, we introduce GAO, a novel game-based framework where a solver and a poser engage in dynamic, competitive interactions. Through this interplay, the poser incrementally generates problems that expose the solver's weaknesses, while the solver continuously adapts its reasoning strategies, producing targeted reinforcement learning data of appropriate difficulty.

Our experiments show that GAO achieves state-of-the-art performance across multiple benchmarks, with particularly notable improvements on more challenging problems. However, analysis indicates that GAO's problem-solving performance relies on longer reasoning chains, which leads to increased resource requirements during inference.

We hope future research can build on GAO by incorporating more efficient reasoning strategies, such as adaptive chain-of-thought pruning or resource-aware inference techniques, enabling the model to maintain high problem-solving performance while keeping computational requirements manageable and scalable for larger benchmarks.

ETHICS STATEMENT

During the data generation process of GAO, although a portion of the generated data was manually reviewed by the authors and the model is intended to be open-sourced, the dataset may still contain content that could be considered harmful or inconsistent with human ethical standards. We encourage users to exercise caution and apply appropriate filtering when utilizing this dataset in future research.

REPRODUCIBILITY STATEMENT

To promote reproducibility and facilitate the review process, we provide data processing scripts at the anonymous link `https://anonymous.4open.science/r/anonymous_test-0D24/` and plan to publicly release the trained models for testing in the future. Additionally, we provide a detailed description of the dataset construction process and experimental settings to further enhance reproducibility. This ensures that other researchers can validate our findings and build upon our work, fostering collaboration and advancing progress in the field.

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

# A CASE STUDY

## A.1 PROMPTS FOR POSER INITIALIZATION

As an experienced education expert, you
need to analyze the most core knowledge
point from a math problem and act as a
question setter to think about how to
build the question step by step from
this knowledge point.
# EXAMPLE
Question: {example_question}
Model Answer: {example_answer}
# NOW
Question: {problem}
Model Answer: <concept>...</concept>
<think>...</think>
(If the question is in English, then
your answer should also be in English.)

Figure 5: Effect of token budget on GAO's chain-of-thought reasoning and problem-solving performance on OlymMATH.

As shown in Figure 5, we utilize the prompt to guide Qwen2.5-72B-Instruct in producing both the underlying problem concepts and their construction process. The construction process is then concatenated with the original problem through simple rules, yielding the answer component used for initializing the poser's SFT training. For clarity and brevity, the manually constructed example question and example answer are not included in the paper, but they were used internally to guide the model during experiments.

## A.2 EXAMPLE OF POSER INPUT AND OUTPUT

Here we provide an example of a poser-generated problem, as illustrated in Table 4. The poser requires only a fixed prefix and a defined knowledge scope to generate a problem. Given this prompt, the poser produces a chain of thought that simulates a mathematical expert incrementally constructing a coherent and appropriately challenging question. As shown in Table 5, the problem-solving reasoning is enclosed between $< think >$ and $< /think >$, while the final generated problem appears after the chain of thought, facilitating straightforward extraction of the problem itself.

| Poser Prompt Example |
| --- |
| As an experienced mathematics education expert, you are now tasked with designing a challenging math problem from scratch, based on a broad area of knowledge. The goal is to thoroughly assess students' abilities. During the problem construction process, you must ensure that the problem is both solvable and sufficiently difficult. The knowledge point is as follows:L'Hôpital's Rule and Taylor Series Expansion in the evaluation of a limit |

Table 4: Input prompt for the poser.

---

**Poser Generated Problem Example**

---

$< think >$ When designing a question that involves the application of L'Hôpital's Rule and Taylor Series Expansion in the evaluation of a limit, as an educational expert, I focus on several key aspects to ensure the problem is both educational and challenging:

1. Mathematical Background and Concept Introduction

Context Selection: The problem is set in the context of limit evaluation, a fundamental topic in calculus. The limit involves an indeterminate form $\frac{0}{0}$ as $x \to 0$, which is a classic scenario where L'Hôpital's Rule can be applied. Incorporating Trigonometric and Logarithmic Functions: The problem includes $\sin x$ and $\ln(1 + x)$, which are common in calculus and provide a good opportunity to explore the behavior of these functions near zero.

2. Parameter Setting and Function Selection

Numerator and Denominator: The numerator is $(\sin x - \ln(1+x))^2$, and the denominator is $x^3 \cdot \tan(2x^3)$. Both the numerator and the denominator approach zero as $x \to 0$, making the limit an indeterminate form $\frac{0}{0}$. Complexity of the Expression: The squared term in the numerator adds a layer of complexity, requiring the student to apply L'Hôpital's Rule more than once or to use series expansions to simplify the expression.

3. Teaching Goals and Skill Development

L'Hôpital's Rule Application: The problem tests the student's ability to apply L'Hôpital's Rule correctly, especially in cases where the rule needs to be applied multiple times. Taylor Series Expansion: The problem also encourages the use of Taylor series expansions, which can simplify the limit evaluation process by approximating the functions involved. Algebraic Manipulation: Students need to manipulate the expression to simplify it, which involves combining and rearranging terms.

4. Step-by-Step Problem Solving

Identify the Indeterminate Form: Recognize that both the numerator and the denominator approach zero as $x \to 0$. Apply L'Hôpital's Rule: Differentiate the numerator and the denominator with respect to $x$ and simplify the resulting expression. Check for Further Indeterminate Form: After applying L'Hôpital's Rule once, if the limit is still in an indeterminate form, apply the rule again. Use Series Expansions: If the limit is still not straightforward, use the Taylor series expansions of $\sin x$ and $\ln(1 + x)$ to approximate the numerator and the denominator. Simplify and Evaluate: Combine the approximations and simplify the expression to find the limit.

5. Educational Objectives and Skill Enhancement

Conceptual Understanding: Students will deepen their understanding of L'Hôpital's Rule and the utility of Taylor series expansions in limit evaluation. Problem-Solving Skills: The problem encourages students to think critically about how to handle complex expressions and apply multiple techniques to reach a solution. Mathematical Rigor: The problem promotes the development of rigorous mathematical reasoning and the ability to handle multiple steps in a problem-solving process.

6. Logical Flow of the Question

Set the Context: Introduce the limit $\lim_{x \to 0} \frac{(\sin x - \ln(1+x))^2}{x^3 \cdot \tan(2x^3)}$. Identify the Indeterminate Form: Show that both the numerator and the denominator approach zero as $x \to 0$. Apply L'Hôpital's Rule: Differentiate the numerator and the denominator and simplify the expression. Check for Further Indeterminate Form: If necessary, apply L'Hôpital's Rule again. Use Series Expansions: If L'Hôpital's Rule is not sufficient, use the Taylor series expansions of $\sin x$ and $\ln(1 + x)$ to approximate the functions. Simplify and Evaluate: Combine the approximations and simplify the expression to find the limit.

By following these steps, the problem aims to challenge students to apply advanced calculus techniques and enhance their problem-solving skills in a structured and educational manner. $< /think >$

Compute the following limit: $\lim_{x \to 0} \frac{(\sin x - \ln(1+x))^2}{x^3 \tan(2x^3)}$

---

Table 5: Example of a poser-generated problem.

# B    ADDITIONAL STATEMENTS

## B.1    THE USE OF LARGE LANGUAGE MODELS (LLMS)

In our paper, LLMs were used during the iterative stage of GAO data synthesis, with specific examples provided in Appendix A. Aside from this, LLMs were only employed for polishing the writing of the paper.

