# OpenReview forum: "Generative Adversarial Optimization: Dual-Reward Reinforcement Learning for Mathematics Reasoning"
_ICLR.cc/2026/Conference — ICLR 2026 Conference Withdrawn Submission_

### Official Review · Reviewer_BHnG · 2025-10-28

**Soundness:** 3
**Presentation:** 3
**Contribution:** 3
**Rating:** 4
**Confidence:** 4

**Summary:**

This paper declares that existing reinforcement learning (RL) frameworks face two challenges:

1. The uncontrollable difficulty of training data, which leads to skewed reward distributions and low learning efficiency.
2. The limited capability ceiling imposed by the initial abilities of large language models (LLMs).

To address these issues, this paper proposes Generative Adversarial Optimization (GAO), a dual-reward RL framework consisting of a poser and a solver. Both are initialized through supervised fine-tuning (SFT) to enhance baseline capabilities, and the GAO algorithm filters problems based on the pass rate so that the poser generates progressively harder tasks while the solver improves by solving them. Experiments demonstrate that GAO achieves state-of-the-art (SOTA) performance on multiple benchmarks compared to previous models of the same size.

**Strengths:**

1. The overall idea of this paper is reasonable. The two challenges indeed exist, and GAO alleviates or addresses them.
2. The experimental results are promising. GAO outperforms existing models of the same size, demonstrating its effectiveness.

**Weaknesses:**

**Ablation study is missing.** The paper mentions an ablation study when listing its contributions, but no corresponding experiments or results are presented later. Despite the overall promising experimental results, an ablation study is still needed to analyze the contribution of each proposed component, for example, the effects of SFT initialization and difficulty-based filtering.

**Dependence on external models.** The GAO framework has an inherent limitation: its strong dependence on external models. The GAO framework relies on consistency-voting reward signals from more powerful LLM judges (Qwen3-235B-A22B and DeepSeek-R1). If these judges contain systematic biases, the solver may end up learning answers that align with "what the judges believe to be correct" rather than absolute mathematical truth, thereby undermining the objectivity of learning.

**Questions:**

1. GAO employs voting from external reasoning experts, known as the LLM-as-a-Judge technique. This inherently imposes an upper bound on reasoning performance, since GAO cannot surpass the reasoning capabilities of its voter models. Consequently, when training larger models with the goal of exceeding all existing models, GAO would likely lose its effectiveness. How can this limitation be addressed?

2. According to Table 1, GAO underperforms Qwen3-14B on GSM8K and MATH500, and its average performance across all benchmarks is only slightly higher than that of Qwen3-14B. This raises the question of whether the performance gains introduced by GAO are substantial compared to the improvements brought by the scaling law.

---

### Official Review · Reviewer_TLo1 · 2025-10-30

**Soundness:** 2
**Presentation:** 2
**Contribution:** 2
**Rating:** 2
**Confidence:** 4

**Summary:**

This paper proposes GAO, an adversarial reinforcement learning framework designed to enhance the mathematical reasoning abilities of LLMs.
GAO trains a Poser that generates challenging problems and a Solver that learns to solve them, enabling both models to co-evolve through iterative competition.

**Strengths:**

This paper introduces GAO, a novel dual-reward reinforcement learning framework that jointly trains a problem poser and a solver through adversarial interaction.
It effectively generates adaptive, challenging mathematical problems that continuously expose the solver’s weaknesses, leading to strong gains in mathematical reasoning without requiring manually annotated data.

**Weaknesses:**

- The paper does not compare the poser’s reasoning performance with the solver trained on poser-generated problems. Without such a comparison, it remains unclear why we cannot simply use the poser itself for reasoning instead of training the solver.
- The paper lacks direct ablation studies, so it is unclear which components of GAO actually contribute to the performance improvements.
- The paper does not quantitatively show how valid the generated problems were, nor how invalid the discarded problems actually were.
- Training both the Solver and the Poser roughly doubles the computational cost compared to standard training.
- The idea of improving performance through competition is not particularly novel.
    - https://arxiv.org/abs/2404.10642
    - https://arxiv.org/abs/2311.08107
    - https://www.arxiv.org/abs/2510.18407
    - https://arxiv.org/abs/2504.19162

**Questions:**

- Qwen2.5-72B-Instruct may have been trained on mathematical datasets. If that is the case, the method indirectly relies on supervised mathematical data, making the “annotation-free” claim overstated.

---

### Official Review · Reviewer_rjqK · 2025-11-01

**Soundness:** 2
**Presentation:** 2
**Contribution:** 2
**Rating:** 2
**Confidence:** 4

**Summary:**

the paper describes a method that uses GAN like optimization called GAO, which iteratively trains a solver and poser, where the poser tries to raise questions the solver cannot solve, and the solver tries to solve the questions. Gain improvement over some baselines on math datasets.

**Strengths:**

1. Improvement over baseline on a collection of datasets
2. The idea of self-training like methods does not rely on data collection

**Weaknesses:**

1. Crispy evaluation against baselines. Key ablations of the poser are missing (e.g. how would it work without filtering (both large model filtering and difficulty filtering)?
2. The method itself relies heavily on forward inference of large scale close models. No evidence that the current poser works better than simple synthetic data based methods, such as completely removing the poser and use deepseek or qwen72B for posing questions and apply filtering similarly, as the inference burden is similar.
3. Only experiments on qwen3, raising concerns of generalization.

**Questions:**

1. Does the current method works better than simple synthetic data generation methods? (make data with deepseek, filter out questions of larger models acc less than 20% or more than 60%, similar as you do)
2. Does it work on other models?
3. How would the performance change as the filter bar changes?

---

### Note · Authors · 2026-01-12

I have read and agree with the venue's withdrawal policy on behalf of myself and my co-authors.